

# Weight change-related factors during the COVID-19 pandemic: a population-based cross-sectional study using social cognitive theory

Roxane Assaf and Jumana Antoun

American University of Beirut, Beirut, Lebanon

Corresponding author
Jumana Antoun, ja46@aub.edu.lb

## ABSTRACT

**Background:** Published studies during the Coronavirus-19 (COVID-19) pandemic have focused on eating and exercise behaviors and failed to portray a comprehensive understanding of the factors associated with weight change in a setting of a behavioral change framework. This study explores factors associated with weight change during the COVID-19 pandemic among Lebanese residents using the Social Cognitive Theory (SCT) framework, integrating behavioral, environmental, and cognitive factors.

**Materials & Methods:** This study uses a cross-sectional design using an anonymous online survey. Participants were recruited from a tertiary hospital patient portal and social media posts. The survey included four domains: demographics, cognitive, behavioral, psychological, and environmental factors. Multiple validated self-reported instruments were included Generalized Anxiety Disorder-2 items (GAD-2), Patient Health Questionnaire-2 (PHQ-2), General Self Efficacy Scale (GSES), Alcohol Use Disorders Identification Test-Concise (AUDIT-C), and the dietary pattern evaluation tool.

**Results:** A sample of 335 complete responses was obtained. Mean age was 39.0 ± 13.4 years old. Participants were mostly females ($n$ = 224, 66.9%), employed ($n$ = 191, 57.4%), nonsmokers ($n$ = 227, 70.5%), reporting depression ($n$ = 224, 80.3%) and anxiety ($n$ = 242, 84.3%). Mean weight change was −7.0 ± 6.0 kg in the decrease weight group and 6.4 ± 5.0 kg in the increase group. When compared to stable weight, the multinomial logistic model factors that were found to correlate significantly to weight gain were: overeating/binge eating ($p$-value = 0.001) and unbalanced food pattern ($p$-value = 0.012). Baseline BMI ($p$-value = 0.003), anxiety ($p$-value = 0.020) and smoking ($p$-value = 0.004) were significant factors of weight loss as compared to stable weight.

**Conclusions:** COVID-19-related weight change is multifactorial and is associated with specific behavior and individual characteristics. Hence, addressing people's behaviors and relationship to food is vital to control weight change during this continuing and future pandemic or natural occurrence.

## INTRODUCTION

Obesity is an emerging global pandemic that impacts both physical and mental health. Its prevalence increased two-fold in the last 40 years, regardless of age, gender, ethnicity, or social status. If current trends continue, half of the world's population will be obese by 2030 (*Mehrzad, 2020*). Obesity is linked to several co-morbidities, including type-2 diabetes, osteoarthritis, cardiovascular disease, and psychological disorders, which have significant economic and health consequences (*Dixon, 2010*). On March 11, 2020, the World Health Organization (WHO) declared the Novel Coronavirus-19 (COVID-19) outbreak a global pandemic (*Cucinotta & Vanelli, 2020*). This pandemic necessitated lockdown, quarantine, and social isolation measures, including remote work and outdoor activity restrictions (*Rehman & Ahmad, 2020*), affecting people's behaviors, weight, and overall health. There was a spectrum of weight change ranging from weight loss (35% of the population), no weight change (42%), to weight gain (23%) (*Deschasaux-Tanguy et al., 2020*; *Di Renzo et al., 2020*; *He et al., 2020*; *Matsungo & Chopera, 2020*; *Scarmozzino & Visioli, 2020*; *Sidor & Rzymski, 2020*; *Zachary et al., 2020*). Whether weight gain or loss, the average weight change was 2 kg during the weeks of lockdown (*Deschasaux-Tanguy et al., 2020*).

Current literature has shed light on some predictors of weight gain during the COVID-19 pandemic. The individual's behavior change could explain weight gain during the COVID-19 pandemic. Individuals who gained weight were more likely to snack (*ALMughamis, AlAsfour & Mehmood, 2020*; *Deschasaux-Tanguy et al., 2020*; *Sidor & Rzymski, 2020*; *Zachary et al., 2020*), eat ready-made food, fast food, meat and dairy products (*Al-Musharaf, 2020*; *Rehman & Ahmad, 2020*; *Scarmozzino & Visioli, 2020*; *Sidor & Rzymski, 2020*), eat fewer fruits and vegetables (*Di Renzo et al., 2020*; *Rehman & Ahmad, 2020*), drink more alcohol (*Cucinotta & Vanelli, 2020*; *He et al., 2020*), smoke more (7, 11, 12) and exercise less (*Cucinotta & Vanelli, 2020*; *He et al., 2020*; *Zhang et al., 2020*). Furthermore, during the COVID-19 lockdown, stress has been linked to weight gain (*Scarmozzino & Visioli, 2020*), with higher levels of depression predisposing to weight gain (*Deschasaux-Tanguy et al., 2020*). Surprisingly, higher anxiety levels were protective against weight gain and linked to weight loss (*Matsungo & Chopera, 2020*). Most importantly, there was a complex interaction between obesity before the pandemic and weight gain during the lockdown. Being of average weight before the pandemic had a higher impact on weight gain during lockdown (*ALMughamis, AlAsfour & Mehmood, 2020*; *Deschasaux-Tanguy et al., 2020*; *He et al., 2020*). One explanation is that people who are overweight or obese have a mentality that is geared toward higher levels of concern with shape and weight, making them more resistant to weight gain during the pandemic (*Haddad et al., 2020*). Finally, female gender and young age (under 35 years old) were associated with weight gain during COVID-19 lockdowns (*Deschasaux-Tanguy et al., 2020*; *He et al., 2020*; *Matsungo & Chopera, 2020*; *Sidor & Rzymski, 2020*).

Identifying weight change-associated factors during the COVID-19 pandemic aids in developing targeted interventions to reverse behavior and prevent health consequences. The current literature has failed to depict a comprehensive understanding of the

predictors, instead focusing on one or two predictors at a time, lacking the context of a framework. A theoretical framework is essential to guide the selection of the associated factors in such a complex multifactorial behavior change. For example, a systematic review of internet-based behavioral interventions has found that interventions that used theory to select participants tended to be more effective (*Webb et al., 2010*). The Social Cognitive Theory (SCT), one of the most extensive dynamic models used for behavior change, describes how people adjust their behavior through control and reinforcement to achieve sustainable goal-directed behavior (*Bandura & Walters, 1977*). The theory highlights the interplay between internal and external social factors, including cognitive, personal, environmental, and behavioral factors. Therefore, this research explores a broader range of factors associated with weight change during the COVID-19 pandemic using the SCT framework based on behavioral, environmental, and cognitive themes. Based on our findings, multiple interventions could be instituted to reduce quarantine's acute and chronic impact on health and weight during this and future pandemics.

## MATERIALS AND METHODS

This study is a quantitative, cross-sectional, survey-based study conducted *via* an anonymous online questionnaire between July 15, 2021, and October 10, 2021. Participants were recruited from the list of American University of Beirut-Medical Center (AUBMC) patients who are active on the electronic medical record patient portal and have shown interest in being involved in research projects. The targeted population received a message notification at their portal. Three reminders were sent weekly. Furthermore, recruitment occurred through a post on AUBMC's and researchers' social media accounts. The following were the inclusion criteria: Lebanese adults at or above 18 years of age residing in Lebanon. Exclusion criteria include individuals under 18 years, women who have been pregnant during the previous year because their weight change may not be related to the COVID-19 pandemic, individuals with a nationality other than Lebanese, and Lebanese individuals residing outside Lebanon.

The survey included four domains: demographics, cognitive, behavioral, psychological, and environmental factors (Supplement 1). Appendix 1 shows the mapping of the various variables to the theoretical framework. Multiple validated tools were included in the survey: Generalized Anxiety Disorder-2 item (GAD-2), Patient Health Questionnaire-2 (PHQ-2), General Self Efficacy Scale (GSES), Alcohol Use Disorders Identification Test-Concise (AUDIT-C), and the dietary pattern evaluation tool.

Both the GAD-2 and PHQ-2 are first-step screening tools for generalized anxiety disorder (GAD) and major depressive disorder (MDD) for their ease of comprehension and administration (*Hughes et al., 2018*; *Kroenke, Spitzer & Williams, 2003*). PHQ-2 questionnaire is one of the most widely validated screening tools for major depressive disorder (MDD), with high sensitivity of 85% and specificity of 95% at a cutoff of three or more. The sensitivity and specificity of GAD are 76% and 81%, respectively.

The GSES evaluates an individual's perceived self-efficacy to foresee mechanisms of handling and adapting to new stressful situations. It is a validated, 10-item questionnaire, scored according to a 4-point Likert scale for each item. A total score is the sum of the 10

items, ranging between 10 and 40, with higher scores indicating higher perceived general efficacy (*Kusurkar, 2013*). It has good scale reliability, with Cronbach's alpha of 0.84 (*Kusurkar, 2013*). It was validated among a similar population of Qatari females (*Crandall, Rahim & Yount, 2016*).

The AUDIT-C is a tool adapted from the original 10-question AUDIT scale. It comprises three questions, scored on a scale of 0–12 (with zero indicating no alcohol use), identifying individuals at risk of hazardous drinking or alcohol use disorder. The cutoff in women is 3 and 4 in men, and the higher the score, the more likely the individual's drinking affects their health (*Reinert & Allen, 2007*). It has good test-retest reliability with inter class correlation coefficient of 0.91.

The dietary pattern evaluation tool is a series of simple, easily administered 34 questions that allow assessment of dietary behavior patterns. It highlights four dietary patterns relevant to the COVID_19 pandemic: high fat/high calorie, overeating/binge, dietary impulse, and unbalanced food intake. The questions were scored on a 1–5 scale (1: not at all; 2: no; 3: average; 4: yes; 5: very much so) with no assigned cutoff (*Do Lee et al., 2016*).

### Patient and public involvement

Neither the patients nor the public was involved in the research design.

### Statistical analysis

Descriptive statistics of the demographics and the various factors were performed using means and standard deviation for continuous variables and percentages for categorical variables. The primary outcome, weight change, was measured using the question "How did your weight change during the COVID-19 pandemic?". It was regrouped into (1) decreased, (2) remained the same, (3) fluctuated, and (4) increased. Multinomial logistic regression was conducted using 10 independent factors (education, BMI before the pandemic, smoking, GAD, dietary impulse pattern, overeating/binge eating, unbalanced food pattern, GSES, exercise, and availability of walking space) and change in weight as the dependent variable. The factors were selected based on the statistical significance of the bivariate analysis. Using G*Power 3.1.9.7 software, a sample size of 142 is needed for logistic regression with 16 factors and an effect size of 0.15, with an alpha level of 0.05 and a power of 80%. For a power of 90%, the sample size needed is 175. SPSS software version 24 was used for data analysis.

## RESULTS

### Personal characteristics and social habits of the participants

A total of 653 participants responded to the survey, out of which 335 had complete responses. Participants were mostly females ($n = 224$, 66.86%), with a mean age 38.93 ± 13.29, were in a relationship ($n = 220$, 65.86%), lived in a major city ($n = 210$, 65.56%), were employed ($n = 191$, 57.35%) and earned >7,000,000 LBP per month ($n = 82$, 33.33%), had a university or post graduate degree ($n = 309$, 92.23%) and had a chronic medical condition ($n = 135$, 40.54%) (Table 1). Almost half of the participants ($n = 130$, 46.93%)

Table 1 Sample characteristics (N = 335).

| | Mean ± SD |
|---|---|
| Age | 38.93 ± 13.29 |
| General self efficacy scale* | 30.70 ± 4.98 |
| Eating scale | 95.70 ± 18.08 |
| | n (%) |
| Gender | |
| Females | 224 (66.86) |
| Males | 111 (33.13) |
| Marital status* | |
| Single | 98 (29.34) |
| Married or in a relationship | 220 (65.86) |
| Widowed or divorced | 16 (4.79) |
| Education | |
| No education/Below high school | 1 (0.29) |
| High school or technical school | 25 (7.46) |
| University degree or post-graduate | 309 (92.23) |
| Location* | |
| Major city | 219 (65.56) |
| Village | 61 (18.26) |
| Suburbs | 54 (16.17) |
| Current work status* | |
| Unemployed | 55 (16.51) |
| Self-employed | 36 (10.81) |
| Employed | 191 (57.35) |
| Student | 31 (9.30) |
| Both employed and a student | 20 (6.0) |
| Salary (in LBP)*[1] | |
| <1,000,000 | 19 (7.72) |
| 1,000,000–2,999,999 | 75 (30.49) |
| 3,000,000–4,999,999 | 36 (14.63) |
| 5,000,000–7,000,000 | 34 (13.82) |
| >7,000,000 | 82 (33.33) |
| Presence of chronic medical conditions* | 135 (40.54) |
| Presence of depression* | 224 (80.29) |
| Presence of anxiety | 242 (84.32) |
| Change in exercise habits* | |
| I increased my daily physical activity | 84 (25.23) |
| I decreased my daily physical activity or now sedentary | 160 (48.04) |
| Stayed the same | 89 (26.72) |
| Hours of sleep* | |
| <5 h | 78 (23.42) |
| 6–7 h | 180 (54.05) |

(Continued)

| Table 1 (continued) | |
|---|---|
| | **Mean ± SD** |
| 8 h and more | 75 (22.50) |
| Smoking* | 95 (29.50) |
| Alcohol intake* | 146 (44.92) |
| Weight change* | |
| Decreased | 96 (28.74) |
| Fluctuated or Stayed the same | 142 (42.51) |
| Increased | 96 (28.74) |

**Notes:**
  * Missing values exist.
  1 The minimum wage in Lebanon during that period was 675,000 LBP (https://www.minimum-wage.org/international/lebanon).

**Table 2 COVID related predictors (N = 335).**

| | | **Mean ± SD n (%)** |
|---|---|---|
| Infected with COVID-19* | Yes | 105 (31.53) |
| | No | 228 (68.46) |
| Needed hospitalization* | Yes | 7 (6.66) |
| | No | 98 (93.33) |
| Experienced fear or anxiety related to the COVID-19 virus* | Always | 70 (21.15) |
| | Very frequently | 103 (31.11) |
| | Occasionally | 94 (28.40) |
| | Rarely | 31 (9.36) |
| | Very rarely | 11 (3.32) |
| | Never | 22 (6.65) |
| Mode of work/studying during lockdowns* | Working or studying from home | 130 (46.93) |
| | Commuting to work or university | 71 (25.63) |
| | A combination of both | 76 (27.43) |

**Note:**
  * Missing values exist.

studied or worked from home, while the rest either commuted to work/university or had a combination of remote work/study and commuting (Table 2).

Most of the participants reported depression ($n = 224$, 80.29%) and anxiety ($n = 242$, 84.32%). The mean GSES score (30.70 ± 4.98) was similar to the international average, with higher scores indicating higher perceived general self-efficacy and coping mechanisms (*Scholz et al., 2002*). In the bivariate analysis, GSES was associated with weight loss ($p$-value = 0.043). Table 1 shows the participants' smoking, alcohol intake, sleep, and exercise habits.

Body mass index (BMI) before (26.01 ± 5.42) and after (26.03 ± 7.687) the COVID-19 lockdown did not change with a mean weight change of −0.84 ± 7.32 kg. Weight change was dispersed almost similarly across the four categories of weight change (decreased, remained the same, fluctuated, and increased). Mean weight change was −7.02 ± 6.03 kg in

the decreased weight group, −1.41 ± 4.84 kg in the group where weight remained the same or fluctuated, and 6.42 ± 5.03 kg in the weight increase group.

## COVID-19 related factors

When assessing COVID-19-related factors, the majority experienced fear or anxiety related to the COVID-19 virus ($n = 267$, 80.66%). Only one-third of the participants got infected with COVID-19 ($n = 105$, 31.53%), out of which only 6.66% needed hospitalization ($n = 7$, 6.66%) (Table 2). Very few searched for information linking COVID and obesity (17.27%) or ways to control weight during the pandemic (11.2%) (Table 3).

Of the few who searched for data linking COVID-19 to weight gain, healthcare professionals were the main source of information ($n = 142$, 42.90%) (Table 3). Nevertheless, the participants recognized the effect of obesity on health overall ($n = 269$, 81.3%) and physical appearance ($n = 275$, 83.33%) and agreed that obesity makes them prone to disease ($n = 316$, 95.76%) (Table 3).

During the pandemic, most individuals ($n = 291$, 87.38%) had home-cooked meals almost always or very frequently available and had access to food during lockdowns (whether through online shopping/delivery services, nearby shops, or distributed by the municipality). Consuming fast foods decreased in a third of the individuals ($n = 117$, 35.13%) (Table 4). Most did not visit neighbors or family members ($n = 210$, 62.87%) and did not attend social gatherings ($n = 257$, 76.94%) during the multiple COVID-19 lockdowns. Finally, they had a space to walk outdoors ($n = 232$, 69.67%).

## Multinomial logistic regression analysis

A multinomial regression analysis was performed to investigate the effect of 16 factors on weight change classified as "decrease in weight," "stayed the same," "fluctuated," and "increased." The predictor variables were tested *a priori* to verify that their correlation to weight change was statistically significant ($p < 0.05$) (Tables 5 and 6). The final list of factors included 10: demographics (smoking, education, BMI before pandemic), psychological (GAD, dietary impulse pattern, overeating/binge eating), cognitive (GSES), behavioral (unbalanced food pattern and exercise), and environmental (availability of walking space). The final model was statistically significant ($p$-value < 0.001) with a −2 log-likelihood of 818.660. Pseudo R-Square Nagelkerke was 0.386; Pearson's goodness of fit shows that the model fits the data well with a $p$-value = 0.080. The reference category was "stayed the same." It is more likely that weight would increase as compared to stay the same if the participant had higher scores on overeating/binge eating (odds ratio = 1.16, 95% CI [1.06–1.27], $p$-value = 0.001), and unbalanced food pattern (odds ratio = 0.84, 95% CI [0.74–0.96], $p$-value = 0.012). It is more likely that the weight would decrease as compared to stay the same if the participant had higher BMI (odds ratio = 1.15, 95% CI [1.05–1.26], $p$-value = 0.003), had higher anxiety scores (odds ratio = 1.36, 95% CI [1.05–1.75], $p$-value = 0.02), and was smoker (odds ratio = 3.23, 95% CI [1.45–7.2], $p$-value = 0.004).

**Table 3  Knowledge and attitude towards obesity during COVID-19 pandemic (N = 335).**

|  |  | n (%) |
| --- | --- | --- |
| Main source of information on COVID-19* | Social media | 37 (11.17) |
|  | Browsing the internet | 104 (31.41) |
|  | TV/Radio | 23 (6.94) |
|  | Newspaper (including online newspapers) | 9 (2.72) |
|  | Colleague | 6 (1.81) |
|  | Health care professionals | 142 (42.90) |
|  | Friends/Family members | 10 (3.02) |
| Search for information linking COVID-19 to overweight or obesity* | Always | 20 (6.06) |
|  | Frequently | 37 (11.21) |
|  | Sometimes | 60 (18.18) |
|  | Seldom | 45 (13.63) |
|  | Never | 168 (50.91) |
| Search for information on how to control weight during the COVID-19 pandemic* | Always | 9 (2.72) |
|  | Frequently | 28 (8.48) |
|  | Sometimes | 58 (17.57) |
|  | Seldom | 51 (15.45) |
|  | Never | 184 (55.776) |
| Concerned about my weight because it may affect my health* | Strongly agree | 173 (52.3) |
|  | Agree | 96 (29.0) |
|  | Neutral | 40 (12.1) |
|  | Disagree | 13 (3.9) |
|  | Strongly agree | 9 (2.7) |
| Concerned about my weight because it affects my physical appearance* | Strongly agree | 169 (51.21) |
|  | Agree | 106 (32.12) |
|  | Neutral | 32 (9.70) |
|  | Disagree | 17 (5.15) |
|  | Strongly agree | 6 (1.81) |
| Weight gain makes individuals prone to disease (heart disease, diabetes, hypertension, etc.)* | Strongly agree | 230 (69.70) |
|  | Agree | 86 (26.06) |
|  | Neutral | 8 (2.42) |
|  | Disagree | 2 (0.60) |
|  | Strongly agree | 4 (1.21) |

Note:
* Missing values exist.

As anxiety was an important factor, an *ad hoc* analysis was performed to understand better the factors associated with anxiety. Higher anxiety scores were positively associated with concerns related to the economy, including fear of being laid off or experiencing poorer personal economy ($p$-value = 0.001), experiencing fear related to the COVID-19 virus ($p$-value = 0.007), having fewer hours of sleep ($p$-value = 0.035) and negatively associated with GSES ($p$-value = 0.023).

**Table 4 Individual behavior and practices (N = 335).**

| | | n (%) |
|---|---|---|
| Availability of home-cooked food* | Always | 190 (57.06) |
| | Very frequently | 101 (30.33) |
| | Occasionally | 33 (9.91) |
| | Rarely | 7 (2.10) |
| | Very rarely | 1 (0.30) |
| | Never | 1 (0.30) |
| Access to food during the periods of lockdown* (more than one answer allowed) | Online shopping/Delivery services | 205 (61.56) |
| | Nearby shops | 231 (69.37) |
| | Municipality distributed food | 9 (2.70) |
| Frequency of consuming/ordering fast food* | Increased | 72 (21.62) |
| | Decreased | 117 (35.13) |
| | Stayed the same | 64 (19.22) |
| | I do not order fast food at all | 80 (24.02) |
| Visit neighbors or family members* | Never | 67 (20.06) |
| | Seldom | 143 (42.81) |
| | Sometimes | 91 (27.24) |
| | Frequently or always | 33 (9.90) |
| Attend social gatherings* | Never | 109 (32.63) |
| | Seldom | 148 (44.31) |
| | Sometimes | 56 (16.77) |
| | Frequently or always | 21 (6.29) |
| Space to walk outdoors* | Yes | 232 (69.7) |
| | No | 101 (30.23) |
| Nearby grocery store* | Yes | 307 (92.19) |
| | No | 26 (7.80) |

**Note:**
* Missing values exist.

# DISCUSSION

This quantitative, cross-sectional online survey-based research aimed to determine a broader range of predictors of weight change during the COVID-19 lockdown in the Lebanese population in the setting of the Social Cognitive Theory framework. The factors included general demographic, behavioral, psychological, environmental, and cognitive factors. Weight change was along a spectrum from weight loss (28.7%), fluctuation (20.4%), stable weight (22.2%), and weight gain (28.7%), similar to what was reported in other studies (*Deschasaux-Tanguy et al., 2020*; *Di Renzo et al., 2020*; *He et al., 2020*; *Matsungo & Chopera, 2020*; *Scarmozzino & Visioli, 2020*; *Sidor & Rzymski, 2020*; *Zachary et al., 2020*). The mean weight change was −7.0 ± 6.0 kg and 6.4 ± 5.0 kg in the decreased and increased weight groups. Multinomial logistic regression had shown that it is more likely to have weight increase, as compared to staying the same, with higher scores on overeating/binge eating and unbalanced food pattern. It is more likely to have weight

**Table 5 Bivariate analysis between weight change and categorical independent variables (chi-square).**

| Weight change | | Decreased | Stayed the same | Fluctuated | Increased | *p*-value |
|---|---|---|---|---|---|---|
| Education | Not educated/Below high school/High school/Technical school | 13 (50.0) | 3 (11.54) | 2 (7.70) | 8 (30.76) | 0.043 |
| | University degree or post graduated | 83 (26.94) | 71 (23.05) | 66 (21.42) | 88 (28.57) | |
| Gender | Female | 64 (28.57) | 50 (22.32) | 50 (22.32) | 60 (26.78) | 0.530 |
| | Male | 32 (29.10) | 24 (21.81) | 18 (16.36) | 36 (32.72) | |
| Mode of work/studying | From home | 19 (28.35) | 11 (16.42) | 13 (19.40) | 24 (35.82) | 0.143 |
| | Commuting | 19 (22.35) | 21 (24.71) | 17 (20.0) | 28 (32.94) | |
| | A combination of both | 36 (28.80) | 34 (27.20) | 31 (24.75) | 24 (19.20) | |
| Seeking information linking COVID to overweight/obesity | Always | 9 (45.0) | 3 (15.0) | 2 (10.0) | 6 (30.0) | 0.413 |
| | Frequently | 8 (21.62) | 9 (24.32) | 7 (18.92) | 13 (35.13) | |
| | Sometimes | 13 (21.67) | 14 (23.33) | 13 (21.67) | 20 (33.33) | |
| | Seldom | 9 (20.45) | 14 (31.81) | 8 (18.18) | 13 (29.54) | |
| | Never | 56 (33.33) | 33 (19.64) | 38 (22.62) | 41 (24.40) | |
| Alcohol | Yes | 42 (28.77) | 35 (24.0) | 32 (21.92) | 37 (25.34) | 0.544 |
| | No | 51 (28.33) | 36 (20.0) | 35 (19.44) | 58 (32.22) | |
| Smoking | Yes | 42 (44.21) | 14 (14.74) | 20 (23.53) | 19 (22.35) | <0.001 |
| | No | 48 (21.24) | 57 (25.22) | 46 (20.35) | 75 (33.18) | |
| Exercise | Increased physical activity | 29 (34.94) | 22 (26.5) | 20 (24.1) | 12 (14.46) | 0.002 |
| | Decreased or now sedentary | 38 (23.75) | 27 (16.87) | 32 (20.0) | 63 (39.37) | |
| | Stayed the same | 27 (30.34) | 25 (28.09) | 16 (17.98) | 21 (23.59) | |
| Close grocery | Yes | 84 (27.45) | 70 (22.87) | 66 (21.57) | 86 (28.10) | 0.189 |
| | No | 10 (38.46) | 4 (15.38) | 2 (7.69) | 10 (38.46) | |
| Availability of home-cooked meals | Always/very frequently | 86 (29.65) | 66 (22.76) | 61 (21.03) | 77 (26.55) | 0.156 |
| | Occasionally | 5 (15.15) | 8 (24.24) | 5 (15.15) | 15 (45.45) | |
| | Rarely/never | 3 (33.3) | 0 (0) | 2 (22.2) | 4 (44.44) | |
| Availability of walking space | Yes | 61 (26.41) | 56 (24.24) | 55 (23.80) | 59 (25.54) | 0.018 |
| | No | 35 (34.65) | 17 (16.83) | 13 (12.87) | 36 (35.64) | |

**Table 6 Bivariate analysis between weight change and continuous independent variables (one-way ANOVA).**

| Weight change | Decreased M ± SD | Stayed the same M ± SD | Fluctuated M ± SD | Increased M ± SD | *p*-value |
|---|---|---|---|---|---|
| Age | 39.25 ± 14.53 | 40.59 ± 12.0 | 36.65 ± 12.99 | 38.95 ± 13.14 | 0.360 |
| GAD score | 4.58 ± 1.71[a] | 3.84 ± 1.34[a] | 4.47 ± 1.93 | 4.51 ± 1.59 | 0.022 |
| PHQ2 score | 4.59 ± 1.93 | 3.96 ± 1.48 | 4.48 ± 1.69 | 4.44 ± 1.83 | 0.126 |
| BMI before pandemic | 27.09 ± 6.69[a] | 24.40 ± 4.76[a] | 26.61 ± 5.34 | 25.90 ± 4.29 | 0.011 |
| Overeating/Binge eating | 21.18 ± 6.35[ac] | 22.40 ± 5.96[d] | 27.33 ± 6.32[bd] | 29.17 ± 5.69[abc] | <0.001 |
| High fat and calories | 31.66 ± 7.47 | 30.35 ± 7.10 | 31.23 ± 7.29 | 33.21 ± 6.44 | 0.066 |
| Unbalanced food patter | 15.34 ± 3.10[cd] | 16.27 ± 3.01[ab] | 15.69 ± 3.08[bd] | 14.96 ± 2.76[ac] | 0.038 |
| Dietary impulse pattern | 21.22 ± 7.75 | 19.44 ± 6.62[a] | 24.81 ± 8.51 | 25.84 ± 7.50[a] | <0.001 |
| GSES score | 29.81 ± 4.93[a] | 32.01 ± 4.82[a] | 30.50 ± 4.72 | 30.687 ± 5.17 | 0.043 |
| Crowding index | 0.93 ± 0.95 | 0.87 ± 1.08 | 0.90 ± 0.67 | 0.86 ± 0.41 | 0.948 |

Note:
[a] *p* < 0.0001; [b] *p* < 0.0001; [c] *p* < 0.0001; [d] *p* < 0.0001.

decrease, as compared to staying the same, with higher baseline BMI, higher anxiety scores, and being a smoker.

Our model has shown that different factors affected weight increase or decrease during the COVID pandemic. Similar to previous studies that have highlighted that weight gain was related to increased eating, especially snacking and fast food, our model has shown that overeating/binge eating and unbalanced food patterns were the most important factors. Nevertheless, most participants had access to local groceries (92.2%) and homemade food (87.4%), and very few ordered fast food (21.6%). Accordingly, eating behaviors were not related to environmental factors, such as access to fast food or groceries while staying home during the multiple COVID-19 lockdowns, but instead may be related to one's own approach to food which is the product of the levels of anxiety and personal attributes of the individual (smoking and baseline BMI). Along the same line, previous studies have highlighted the protective effect of anxiety and baseline BMI against weight gain (*ALMughamis, AlAsfour & Mehmood, 2020*; *Deschasaux-Tanguy et al., 2020*; *Haddad et al., 2020*; *He et al., 2020*; *Matsungo & Chopera, 2020*). As probably expected during a pandemic, physical activity decreased in half of the sampled population, although most participants had a space to walk. The physical activity decrease was not distinct from the literature (*Alqurashi, 2021*; *Sooriyaarachchi et al., 2021*; *Yang et al., 2021*). In a similar sample in Sri Lanka, a significant decrease in physical activity and increased sitting and screen time were noted, along with one-third of the sample gaining weight (*Sooriyaarachchi et al., 2021*). An important consideration here is that even though environmental factors did not directly correlate to weight change, however, the COVID pandemic had its impact on the individual's anxiety regarding their fear related to the COVID-19 virus and concerns related to the economy, including fear of being laid off or experiencing poorer personal economy.

In our study, baseline BMI, overeating or binge eating, unbalanced food pattern, anxiety, and smoking were the main factors of weight change. We hypothesize that weight gain during the COVID-19 pandemic was triggered intrinsically by one's relationship with food and emotional eating at times of crisis and stress, including natural disasters and eminent fear of infection, such as during the COVID-19 lockdown. For example, different stressful situations have been found to manifest in increased eating, such as the observance of overeating accompanying high levels of distress post-earthquake (*Kuijer & Boyce, 2012*). Emotional eating is highly associated with the development of obesity, notably during stressful periods (*Al-Musharaf, 2020*; *Konttinen, 2020*). In our study, emotional eating was addressed in the form of eight questions entitled "Dietary Impulse Pattern" in the setting of the eating scale. GSES was associated with weight loss in the bivariate analysis ($p$-value = 0.043) and could be interpreted as the more control one has over their impulsivity to eat and the more refined ability to cope with various stimuli, the better control over weight change.

An intuitive question is whether emotional eating is inherited or acquired (*Herle, Fildes & Llewellyn, 2018*). Whether it is one or the other does not truly matter because we, as humans, are the product of both our nature and nurture. Still, the latter is the only entity we can control to alleviate the consequences of obesity in this current pandemic and in the

future (*Freitas et al., 2018*). It was evident that the environment and demographic factors did not help one gain or lose weight during the COVID-19 lockdown; however, behavioral factors played an important role along with psychological factors such as anxiety. Weight seems closely related to one's mental state, general self-efficacy, and eating patterns. Luckily behavior and mental health are entities that could be addressed and, as such, alleviate the effect of the COVID-19 lockdown on weight. Following our results, behavioral support was a major key in weight control during the COVID-19 pandemic (*Zaman et al., 2022*).

The strength of this research is that it was guided by a theoretical model that includes the interplay between demographic, behavioral, environmental, and cognitive factors. Previously published studies focused mainly on weight gain and its associated factors during COVID-19 lockdown rather than the whole spectrum ranging from weight loss to weight gain (*Hammouri et al., 2022*; *Khubchandani et al., 2022*; *Seal et al., 2022*; *Tan, Tan & Tan, 2022*). When compiled in the setting of a framework, the factors that drive weight gain are overeating, binge eating, and having an unbalanced food pattern which are all behavioral factors well known to cause weight gain and can, fortunately, be controlled. On the other hand, our study has uncovered factors associated with weight loss, including baseline BMI before the lockdowns, anxiety, and smoking.

Nevertheless, there are limitations to this study. First, the response rate could not be calculated as the participants recruited through the AUBMC health portal, MyChart, and social media is unknown, which may affect the external validity and generalization of the results. Second, the sample encompasses mostly females of high educational and socioeconomic status, mainly residing in two areas of Lebanon (Beirut and Keserwan), a pattern that occurred in similar studies affecting the generalizability of the results. Furthermore, only Lebanese individuals were recruited; hence specific cultural issues could be different in people of different nationalities (*Alqurashi, 2021*; *Sooriyaarachchi et al., 2021*; *Yang et al., 2021*). Third, the survey included self-reported data with inherent recall bias as individuals were asked to recall information after the lockdown had been withdrawn. They may have referred to the current moment to answer the questions regarding their emotions and behaviors.

Moreover, the survey relied on the self-report of the participants. Finally, the findings of this research only show an association between weight change and the associated factors. It cannot provide evidence of causality. Nevertheless, this study helps establish preliminary evidence that can be used to plan future studies when a cohort of participants is followed throughout the lockdown. Approaching participants through a systematic random sampling of neighborhoods may eliminate the selection bias.

## CONCLUSIONS

COVID-19-related weight change is complex and multifactorial. The COVID pandemic has led to a spectrum of weight changes, with different factors affecting weight gain or loss. Environmental factors related to access to food and walking did not play a major role but instead eating patterns, anxiety, baseline BMI and smoking. Further research is warranted, preferably as qualitative studies, to uncover other factors contributing to the weight

change. In future periods of lockdown, it is strongly advisable to spread more education on emotional eating and anxiety reduction through awareness campaigns to help address and possibly control two of the most important driving forces of weight change. This may prepare individuals not only for future COVID-19 lockdowns but maybe other future pandemics and stressors, including other natural occurrences.

### Funding
The authors received no funding for this work.

### Competing Interests
Jumana Antoun is an Academic Editor for PeerJ.

### Author Contributions
- Roxane Assaf conceived and designed the experiments, performed the experiments, analyzed the data, prepared figures and/or tables, authored or reviewed drafts of the article, and approved the final draft.
- Jumana Antoun conceived and designed the experiments, analyzed the data, prepared figures and/or tables, authored or reviewed drafts of the article, and approved the final draft.

### Human Ethics
The following information was supplied relating to ethical approvals (*i.e.*, approving body and any reference numbers):

The Institutional Review Board of the American University of Beirut granted ethical approval (SBS-2021-0075).

### Data Availability
The raw data is available in the Supplemental Files.

### Supplemental Information
Supplemental information for this article can be found online at http://dx.doi.org/10.7717/peerj.13829#supplemental-information.

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
