# Peer review of "Weight change-related factors during the COVID-19 pandemic: a population-based cross-sectional study using social cognitive theory"

_PeerJ, doi:10.7717/peerj.13829_

## Round 0.1 · original submission · Minor Revisions

I have now received the reviewers' comments on your manuscript. They have suggested some minor revisions to your manuscript. Therefore, I invite you to respond to the reviewers' comments and revise your manuscript.

Reviewer 1 ·

Basic reporting

Abstract:
In the methodology section of the abstract, indicate the number of sample people, sampling method, statistical method, software and confidence interval.
Introduction:
Do you mean by "weight neutrality" the same as not changing weight?
The contents of the introduction are very complete, good and concise. That this cohesion and integrity is commendable. Only in the part where the theory SCT is presented, a kind of disconnection is seen. Reinforce this section and enter into the discussion of the need to use a theoretical framework with a broader explanation. And then introduce the theorist of this intellectual framework and explain this framework well.
Introduce the variables derived from the research by mentioning the source explicitly under the heading of each aspect of this phenomenon

Experimental design

Methodology:
Specify the sampling method.
Questionnaires should be explained separately in each paragraph. Mention the validity and reliability of each questionnaire. Determine which of the questionnaires measures each of the cognitive, behavioral and environmental dimensions of this phenomenon?And give a brief explanation of why you chose these questionnaires and not other questionnaires.

Validity of the findings

Results:
How did you come up with the final list of weight change predictors?
Discussion:
Based on the discussion, it seems that according to the research literature, you have considered several variables to be effective in weight changes by default. And you have chosen the questionnaires based on that and finally in the statistical analysis you have examined the variables. You should explain this process. And prepare the reader's mind.

Additional comments

no comment.

Reviewer 2 ·

Basic reporting

Excellent basic reporting. Clear and concise writing.
Professional and proficient standard of written English that communicated their points and arguments well.
References and background text were well provided
Studies accurately referenced

Experimental design

Yes the primary research is within the aims and scope of journal
Research question is clearly defined and is relevant
Intention is also identified; which is to present a more comprehensive look into the predictors of weight changes during the pandemic

Study design is a simple quantitative cross sectional survey.It would have been more meaningful if it had been a mixed method study with components of qualitative interview or questions. Otherwise method suffices for the the cross sectional study.

Perhaps questions to changes of their eating and sleeping behaviours could be explored too. They had only asked for changes in exercise habits

Validity of the findings

Discussion and conclusion were adequate.

Reviewer 3 ·

Basic reporting

The presented paper is quite interesting and provides inspiration for other researchers to conduct further research. Research reports are submitted in good English writing and are consistent with professional standards of politeness so that they are easy to understand scientifically.

Experimental design

The research carried out is quite identical to the scientific field required by the journal. The literacy used in the literature review assessment can be said to be quite relevant to the objectives of the proposed research. The method is explained in enough detail and is informative to inspire ideas for further research development or models that can be imitated by other researchers.

Validity of the findings

The results of the study are explained in full by including the statistical value of their significance. Supporting data on the research results are presented in detail and clearly.
The discussion is quite specific, targeting the discussion that leads to a comparison of the results obtained from the study compared to other similar studies.
Please improve the way of writing the display in the attached table which is still in closed tabular form. Writing tables according to journal writing guidelines should be written in an open table way.
Attention to table 5 should be displayed in a simpler and more attractive format that focuses on factors that are the focus of attention or are meaningful in discussions and discussions. The same thing is also addressed for table 6 to be summarized so that it is interesting to read.

Additional comments

The title seems unattractive to the reader, it is necessary to add words regarding specific information related to the research.

---

## Round 0.2 · Minor Revisions

Many thanks for the updated paper. It is much better after the changes. However, I cannot recommend publication at this stage for the following reasons:

GENERAL REVIEW

1. Regardless of the results obtained from the model, it is crucial to emphasize that accurate predictions cannot be guaranteed by cross-sectional study. Rather, development of prediction models is based on cohort study. Thus, prediction models resulting from cross-sectional designs can be misleading. Therefore, it is necessary to consider this point in the interpretation of the results of this study. I recommend that the authors use the word association (or relationship) instead of prediction in the whole article. In this regard, I suggest that the title of the article be changed as follows:
“Weight change-related factors during the COVID-19 pandemic: a population-based cross-sectional study using social cognitive theory”

2. Authors should also mention this concern in study limitations. Cross-sectional studies mostly fail to specify a definite reason behind a correlation. In detail, this limitation can prevent a deep understanding of the nature of the causal relationship between study variables. As another limitation, the use of self-report measures may only reflect the feelings of patients during the assessment and not reveal the real emotions they have suffered.

3. The format and structure of the manuscript (i.e., abstract, main text, and references) should be also improved as is mentioned in the author guideline for manuscript submission.

4. Finally, I suggest that the manuscript is proofread by a fluent English speaker or Editing service. So please make sure there are no English errors.

ABSTRACT

1. In this section, the methodology can be written better. For this purpose, it is suggested that the sample size determination and power analysis using G*power software be transferred to the materials and methods section and in the abstract section, the authors try to address the study variables and how to measure them through self-report questionnaires.

RESULTS

1. Please review and correct the results and tables carefully. Please round the data (except p-values) to 2 decimal places in all sections of the article.

DISCUSSION

1. When addressing limitations, your future directions should be tailored to the limitations identified. For each limitation, there should be a future direction that addresses it.

---

## Round 0.3 · accepted · Accept

Many thanks for addressing my concerns.